# A Modularized IoT Monitoring System with Edge-Computing for Aquaponics

**DOI:** 10.3390/s22239260

**Published:** 2022-11-28

**Authors:** Shiqi Wan, Kexin Zhao, Zhongling Lu, Jianke Li, Tiangang Lu, Haihua Wang

**Affiliations:** 1College of Information and Electrical Engineering, China Agricultural University, Beijing 100083, China; 2National Digital Fisheries Innovation Center, Beijing 100083, China; 3College of Information Technology, Hebei University of Economics and Business, Shijiazhuang 050062, China; 4Information Center, Beijing Municipal Bureau of Agriculture and Rural Development, Beijing 100101, China

**Keywords:** aquaponics, internet of things, edge-computing, extensible modules, environmental monitoring

## Abstract

Aquaponics is a green and efficient agricultural production model that combines aquaculture and vegetable cultivation. It is worth looking into optimizing the proportion of fish and plants to improve the quality and yield. However, there is little non-destructive monitoring of plant growth in aquaponics monitoring systems currently. In this paper, based on the Internet of Things technologies, a monitoring system is designed with miniaturization, modularization, and low-cost features for cultivation-breeding ratio research. The system can realize remote monitoring and intelligent control of parameters needed to keep fish and plants under optimal conditions. First, a 32-bit chip is used as the Microcontroller Unit to develop the intelligent sensing unit, which can realize 16 different data acquisitions as stand-alone extensible modules. Second, to achieve plant data acquisition and upload, the Raspberry Pi embedded with image processing algorithms is introduced to realize edge-computing. Finally, all the collected data is stored in the Ali-cloud through Wi-Fi and a WeChat Mini Program is designed to display data and control devices. The results show that there is no packet loss within 90 m for wireless transmission, and the error rate of environment parameters is limited to 5%. It was proven that the system is intelligent, flexible, low-cost, and stable which is suitable for small-scale aquaponics well.

## 1. Introduction

The prototype of aquaponics came from fish farming in rice paddies which happened in ancient China [1]. Modern aquaponics systems integrate aquaculture, soilless cultivation, and modern agricultural information technology [2]. Ibtissame E. et al. [3] compared aquaponics with the hydroponic and the traditional method, pointing out the advantages of aquaponics systems in agricultural production. Aquaponics provides high-quality agricultural products while reducing the ecological impact. With the rapid development of science and technology, agricultural informatization has moved from traditional agriculture to smart agriculture mainly supported by Internet of Things technology [4]. The smart factory of aquaponics was proposed in recent years, which uses smart sensing, information control, and other Internet of Things (IoT) technology to achieve a low-cost, green cycle production model.

While aquaponic farming is an alternative agricultural method [5], the varying adaptability of species to the environment poses significant challenges for research in both academic and commercial sectors. To improve production efficiency as much as possible, it becomes indispensable to monitor and regulate the growth environment of fish and vegetables by using IoT technology. This technology when used alongside innovative agricultural techniques such as aquaponics can lead to new dimensions of food production [6]. Numerous technological studies have been undertaken in an effort to better handle this type of agricultural technology. Studies have shown that it is efficient in aquaponics systems to use intelligent analysis and processing [7]. 

Nhan Chi Nguyen et al. [8] introduced an aquaponics monitoring and control system based on IoT technology, which allows users to monitor the potential of hydrogen (pH) value, Dissolved Oxygen (DO), temperature, and humidity through an application on their smartphones, and to set thresholds to control water and air pumps. This method eliminates manual intervention and simplifies production costs compared with manual monitoring methods, but the monitoring parameters cannot yet be comprehended. Sun Jian et al. [9] conducted a study on an aquaponics environmental monitoring system based on IoT technology and described its role in promoting aquaponics technology. Ma Zichao et al. [10] designed a low-power monitoring device with ZigBee for an aquaponics system to collect temperature and pH values online. The device can be used to remotely control fish feeding, and save water resources effectively. FL Valiente et al. [11] monitored and controlled pH and temperature in a Nile tilapia-lettuce aquaponics system through IoT. The growth of plants and fish in the automated aquaponics system was significantly better than in the traditional aquaponics system. Siyao An et al. [12] designed a data collection device for aquaponics systems to realize real-time monitoring of various environmental data. Then, an aquaponics data service platform was developed to make the best decisions by monitoring the status of the system online. Mohanad Odema et al. [13] proposed a system with Modbus Transmission Control Protocol (TCP) as the primary communication protocol, allowing it to benefit from the advantages of Modbus and TCP technologies simultaneously. Murad et al. [14] have developed a low-cost system for monitoring and controlling water parameters using Arduino as a Microcontroller Unit (MCU). Mandap et al. [15] equipped Arduino with a web application to design an automated aquaponics system that allows the system to control actuators in response to pH level, temperature, and DO parameter changes. However, the systems mentioned above are mainly designed for the monitoring of pH, temperature, and DO, which belongs to water quality monitoring. Some of them have added remote control functions. The data monitoring is not comprehensive enough for aquaponics scenarios.

Further, Haryanto et al. [16] designed a smart aquaponics system that could control and monitor the degree of acidity, water level, water temperature, and fish feed that were integrated with an internet-based mobile application. Mohammad Alselek et al. [17] developed a complete 5G-enabled IoT system for fully monitoring the performance of fishery farms, which improves the state-of-the-art in terms of aquaponics life cycle monitoring metrics and communication technologies. Mehra et al. [18] introduced deep neural networks to the analysis of tomato hydroponic growth parameters and compared them with traditional soil culture. These studies extend the parameters monitored by the system but still did not overcome the problem of automatic acquisition of plant growth condition data. 

In conclusion, up to now, the research on aquaponics systems was mostly focused on system integration, water-quality, environmental parameters monitoring, and fish behavior detection, but many disadvantages have not considered, such as poor mobility, long development cycle, complicated operation, and high power-consumption. Additionally, there is a relative lack of attention to research on plant growth monitoring in aquaponics systems. The image processing technology is a low-cost solution for vegetable growth monitoring. However, for cloud-storage IoT systems, image data upload and cloud computing are limited by bandwidth and transmission delay.

According to the characteristics of aquaponics systems and urban agriculture scenarios, a miniaturized intelligent aquaponics monitoring system is proposed in this paper. This system can realize real-time monitoring of key indicators, such as water quality, planting environment, and plant growth conditions based on intelligent embedded technology and edge-computing. To overcome the impact of uploading speed and delay during processing a large amount of image data, an image edge-computing unit is introduced to the aquaponics system for monitoring growth conditions, such as plant height, plant stem, etc., which can improve the data transmission efficiency rapidly. 

The contributions of our work are three-fold:We propose a modularized IoT monitoring system with edge-computing for aquaponics, which overcomes many problems of traditional handheld aquaponics systems, such as high latency, labor-intensive, low efficiency, and poor scalability.We build an end-edge-cloud system architecture. Using Raspberry PI as an edge sensor, an edge image processing module is implemented, which enables the system to monitor plant growth conditions by deep learning without destruction.We develop a WeChat Mini Program for the monitoring system, improving system management efficiency, and reducing cost. With the software platform, users can easily realize remote monitoring and control of the aquaponics system by smart cell phone.

This paper has the following structure and organization: First, in this section, cases and work related to traditional and modern aquaponics systems are discussed. Section 2 describes the overall architecture and functions of the monitoring system that we designed. The main hardware modules include intelligent sensing unit design, edge image processing module design, and so on are shown in Section 3. Section 4 presents the system software design. The tests and results are listed in Section 5. In Section 6, we summarize our research results, clarify the significance of the work, and analyze the prospects for future work.

## 2. System Architecture

When it involves high yields and quality, designing and managing a system for aquaponic is a difficult task. Due to a combination of a greenhouse and a symbiotic environment, the parameters and factors that must be controlled are diverse. [19]. Therefore, in light of the investigation and analysis of current research related to the aquaponics system, the aquaponics monitoring system is built based on embedded edge-computing and IoT technology to reach the full supervision and abnormal monitoring of water quality (temperature, pH, DO, etc.), planting environment (temperature, humidity, light intensity, etc.), and plant growth conditions (plant height, plant stem, leaf area, etc.). The general framework of the system was presented in Figure 1. 

The IoT can empower systems and machines to communicate with each other and make decisions based on data without human intervention. In general, the architecture of IoT consists of three layers: Perception Layer (sensing), Network Layer (data transmission), and Application Layer (data storage and manipulation) [20]. In this paper, the perception layer mainly contains three parts: the environment sensing unit, the plant growth conditions sensing unit, and the intelligent control unit. The environment sensing unit used Wireless Sensor Network (WSN) technology to realize the collection of water quality and planting environment in the aquaponics system. The edge-computing module with image processing algorithms is introduced to the plant growth sensing unit, which can quickly obtain and upload the indicator data of plant growth and alleviate the network transmission load. Intelligent control unit enables remote control of equipment in aquaponics systems. The network layer uses Wireless Fidelity technology (Wi-Fi) to transmit the processed data to the server for storage via TCP-based Message Queuing Telemetry Transport (MQTT) protocol. The application layer includes the Ali-cloud server and WeChat Mini Program. MySQL database is deployed in the server for cloud storage, sending, and receiving of IoT data. The Mini Program can realize the functions of real-time monitoring, history data display, abnormal alarm, and remote control with the collected data.

## 3. System Hardware Design

### 3.1. Intelligent Sensing Unit Design

The overall framework of the hardware is shown in Figure 2. The perception layer uses the STMicroelectronics 32-bit family of microcontroller chip (STM32) as the MCU, whose periphery includes five major parts: fishpond water-quality monitoring module, planting environment monitoring module, plant growth conditions monitoring module, intelligent control module, and edge-computing module unit; the network layer uses ESP8266 for Wi-Fi wireless communication. 

In Figure 2, the water-quality monitoring module includes pH sensors, water temperature sensors, DO sensors, signal amplification, and peripheral signal conversion circuits. The planting environment monitoring module includes temperature-humidity sensors, CO_2_ sensors, light intensity sensors, and peripheral signal conversion circuits. The intelligent control module can remotely start/stop the pump, light, and alarm to accomplish the intelligent regulation of the system. The edge-computing unit is designed to perform localized calculations on image data for efficient acquisition of growth conditions data, realizing the continuous monitoring of the plant stem and plant height. 

### 3.2. Sensor Selection

The small-scale aquaponics system mainly demonstrates the intelligent monitoring of three key indicators of aquaponics: DO, light intensity, and plant height. For other needs, users can also connect the required sensors to Input/Output (I/O) ports for real-time monitoring of more indicators. 

Two types of DO measurement principles are commonly used: electrochemical electrodes and polarographic electrodes. In this paper, electrochemical electrodes were chosen because of their fast start-up and low-maintenance features. Moreover, the XC6206 voltage regulator and TP5551 zero-drift operational amplifier are used to design the signal conditioning module peripherally. The BH1750 is a digital light intensity sensor integrated chip with an input light range of 1-65535lx and a minimum error variation of ±20%. It mainly uses the photovoltaic effect generated by the internal photodiode to convert the light signal into an electrical signal for calculation. The Raspberry Pi camera module (Pi Cam) was released in May 2013 and is equipped with a 5 Mpixel sensor that can be connected to the Camera Serial Interface (CSI) on the Raspberry Pi easily. Considering that the small-scale aquaponics system is currently under exploration, part of sensing devices in the multi-sensing process, such as DO, light intensity, and CO_2_, are selected for system verification in this paper. The specific sensor selection results are listed in Table 1.

### 3.3. Edge Image Processing Module Design

Due to the limitation of serial port speed and data processing capability of the STM32, it is difficult to transmit and process image data remotely in an instant. Therefore, to deal with this problem, this paper integrated the image processing module based on Raspberry Pi for collecting growth conditions into the perception layer as the edge-computing device. Thus, the end-edge-cloud system architecture is constructed, and the edge image processing module is extended with flexible image data. A lightweight training model is installed on Raspberry Pi with the Pytorch framework. The deep learning algorithm is integrated to capture plant growth conditions within 10 s by performing target detection, target cropping, image enhancement, and binarization calculations on image data. In other words, the extraction of plant growth condition data can be done quickly, precisely, and non-destructively by shooting on site. The above processing results will be transmitted to the cloud storage through wireless communication. The structure of the edge image processing module is shown in Figure 3.

In this paper, the edge image processing architecture is based on the concept of edge-computing. According to the Edge-computing Reference Architecture 3.0 (2018) [21], its basic architecture is shown in Figure 4. Edge-computing refers to processing the data on the device and very close to the device [22], which has better real-time monitoring, stability, and security compared to traditional cloud computing and can be proposed to surmount the problems caused by high latency and low bandwidth. The edge image processing module will complete the image data acquisition and analysis calculations in advance. STM32 is responsible for the automatic transfer of packaged data to the cloud. Accordingly, not only is the pressure of data explosion and stream relieved greatly but also the efficiency of information transmission is improved [23].

### 3.4. Data Acquisition Circuit Design

The sensor detection module consists of four parts: voltage stabilization, signal amplification, voltage following, and temperature compensation. The internal resistance of some composite electrodes is extremely high, for example, up to 10^12^ Ω for glass electrodes. Since a high impedance input is required, the correct voltage signal can be obtained only when the input impedance of the op-amp is higher than the internal resistance of the sensor. The design uses a TLC4502 dual Operational Amplifier (Op-Amp) with an input impedance of 10^12^ Ω at room temperature to realize the function of signal amplification and convert microvolts to voltages in the range of 0 to 5 V, which can be sent to the main control chip for data processing. In addition, the temperature is an important factor that affects the sensor output voltage, so DS18B20 is used as a temperature compensation sub-module to improve the data accuracy. The analog to digital converter (ADC) circuit is mainly used to convert the analog signals collected in real-time into digital signals that can be processed by the MCU to support its processing of the signals collected in real-time. The schematic diagram of the data acquisition circuit is shown in Figure 5.

### 3.5. Main Control Circuit and Peripheral Circuit Design

The main control circuit and peripheral interaction circuit are presented in Figure 6. To match low power and multi-signal scenarios, the MCU uses STM32F103C8T6 as the main control chip, which is used to provide control signals for data monitoring and transmission circuit. It is a 32-bit microcontroller based on the Cortex-M3 core introduced by STMicroelectronics (ST), with high performance, low cost, and low power consumption. In addition, it integrates various communication serial ports, with two Inter-Integrated Circuit Bus (IIC) interfaces, two Serial Peripheral Interfaces (SPI), three Universal Synchronous/Asynchronous Receiver/Transmitter (USART), and one Controller Area Network (CAN), and a total of twelve channels of ADC, which facilitates easy development of different data requirements. Peripheral circuit includes screen display circuit, alarm circuit, and power switching circuit, which are connected through the I/O port for real-time display of data monitoring and chirp alarm when the data is abnormal. Organic Light-Emitting Diode (OLED) driven by SSD1306; buzzer driven by S8050.

### 3.6. Wireless Transmission Network Design

The network layer is one of the key layers in the system, as a bridge between the perception layer and the application layer, whose main task is to accomplish the interconnection of the information collected by the underlying sensors and the upper layer applications. The wireless transmission unit adopts ESP8266 for Wi-Fi transmission, which is a low-power, cost-effective embedded wireless network module. The complete TCP/IP protocol stack has been embedded in the module, so it can send data from traditional serial devices to the Internet. It supports three working modes: serial to STA, serial to AP, and STA + AP, and supports the standard IEEE802.11 b/g/n protocol. The features of the module cover no new wires, easy to use, quick start, etc. The serial port rate can reach up to 4 Mbp. Figure 7 shows the material object of the data acquisition node and transceiver.

## 4. System Software Design

### 4.1. Intelligent Sensing Unit Program Design

The hardware part was developed in Keil and programmed in C, which mainly implements the sensing data acquisition and Wi-Fi wireless transmission functions. The intelligent sensing unit mainly has three major states: initialization, status judgment, and data reading. The block program diagram of the intelligent sensing unit is shown in Figure 8.

After each module is initialized in turn, the STM32-ESP8266 connection request will be sent out. Wait for 10 s for the server reply after a successful connection, the buzzer will beep to indicate successful access. Otherwise, it will reset the ESP8266 for reconnection. 

With the PB6 pin going high, the IIC is started during the falling of edge PB7 pin. Then the MCU sends power-on and measurement commands to the IIC device. The measurement ends when the ACK signal of BH1750 is detected, and the calculation of light intensity is performed after reading the register data. Similarly, when the PB10 pin goes high, the PB11 pin shows a falling of edge to indicate the IIC is woken up, and SGP30 reads and writes CO_2_ data in sequence. 

After ADC initialization, ADC conversion is triggered using software to calibrate the voltage and data conversion, then the DO value is calculated by formula. After the data measurement, the data display work is performed by timer TIM2. The status of the PA0 pin is read as a flag to determine whether there is data abnormality and alarm action. 

The program for the edge-computing module was developed in Raspberry Pi using Pycharm and programmed in Python. The main function is to provide acquisition commands for pictures of plants during growth and integrate segmentation and detection algorithms to achieve segmentation of plant targets and calculation of growth indicators and transfer the result data to the main control module for cloud storage.

### 4.2. Wireless Transmission Network Program Design

The network layer uses Wi-Fi. The ESP8266 modules supporting the three modes are used to send the data measured by the smart sensing unit to the cloud server for database storage via TCP-based MQTT protocol, which is convenient for historical data queries and later relationship analysis. In this design, the communication module is set to STA mode using the AT command, when the device is connected to the AP hotspot as the station. Table 2 represents the specific configuration of part of the command.

UART2 is configured as a communication serial port, with TXD pin connected to the PA2 pin and RXD pin connected to the PA3 pin. Turn on timer TIM3, set 10 ms interruption, and the program will determine whether the data is continuous by calculating whether the time difference between 2 consecutive characters is greater than 10 ms. The circular queue is used as the buffer for data reception, which is used to save the data received in the UART2 interrupt function. Signal REV_OK indicates that the reception is complete, and the data can be sent out. The block diagram of the wireless transmission network program is shown in Figure 9.

### 4.3. Application Layer Platform Design

#### 4.3.1. IoT Gateway Design

Elastic compute service (ECS) is a simple, efficient, safe, and reliable computing service with elastic and scalable processing capacity. At present, the larger domestic cloud service platforms include Ali-cloud, Baidu Cloud, Tencent Cloud, WeChat hardware platform, Giz Cloud, OneNET, etc. Among them, Ali-cloud has the characteristics of high stability, full functionality, and flexible development, which is an infrastructure as a service (IaaS) cloud server with superior performance. In this paper, we choose Ali-cloud as the storage and processing center of cloud data and use Community Enterprise Operating System (CentOS) 7.2.1 to build and configure the server. The overall architecture of the application layer platform is shown in Figure 10.

The server mainly includes two parts: remote data reception and database management. The water-quality data and environmental data collected by the intelligent sensing unit use JavaScript Object Notation (JSON) format to communicate with the remote server, and the server will subscribe to a topic for port listening. The design uses MySQL, which is fast and flexible, as the database for cloud storage and management. Through the programming interface it provides, the master control module is able to upload the collected data to Ali-cloud through Wi-Fi and able to receive cloud server commands to achieve interconnection between services and devices. 

#### 4.3.2. Mini Program Design

In this paper, Visual Studio with the mpVue framework was used for WeChat Mini Program development, including four parts: user login, real-time monitoring, history display, and remote control. Figure 11 represents the framework of the Mini Program. The most important feature is that it can be accessed without downloading. The Mini Program realizes the user experience of “come and use”, which is one of the best alternatives to mobile client apps. 

According to the registration information, users can log in to check the real-time data of the aquaponics system. The Mini Program will automatically send a server connection request. With a successful connection, the Mini Program gets real-time data through the subscribed “PubTopic” and sends control commands through “SubTopic”. Finally, it realizes the display of weather overview, water quality, planting environment, and plant growth conditions on the monitoring page and the switch on and off control of water pump, light, and alarm on the remote-control page, which can assist to complete planting evaluation with the mature model deployed in the cloud.

## 5. Experiments, Results and Discussion

### 5.1. Experiments Setting and Evaluation Indices

Three experiments were conducted in this paper, which correspond to verifying the transmission reliability, system stability, and measurement accuracy respectively. In the data packet loss rate test, the packet loss rate (PLR) at different distances (DIS) was used as the evaluation index. The calculation method is shown in Equation (1), NOP stands for the number of packages and NORP stands for the number of receiving packages. Low PLR means system transmission is reliable. The stability test is primarily concerned with whether the system can be online for a sustained period without going offline. The field tests mainly verified the accuracy of the system measurements by comparing the manual sampling (actual values) with the system recording values. Mean absolute error (MAE), mean squared error (MSE), and root mean square error (RMSE) were used as evaluation indices in this paper. The calculation method is shown in Equations (2)–(4), $y_{i}$ is the actual value measured manually, and $\hat{y_{i}}$ is the predicted value recorded by the system. Similarly, the lower these indices are, the smaller error of the system measurement is.
(1)$$\mathrm{PLR}=1-\frac{\mathrm{NORP}}{\mathrm{NOP}}$$
(2)$$\mathrm{MAE}=\frac{1}{m}\overset{m}{\underset{i=1}{\sum }}\left|\left(y_{i}-\hat{y_{i}}\right)\right|\;$$
(3)$$\;\mathrm{MSE}=\frac{1}{m}\overset{m}{\underset{i=1}{\sum }}{\left(y_{i}-\hat{y_{i}}\right)}^{2}$$
(4)$$\mathrm{RMSE}=\sqrt{\frac{1}{m}\overset{m}{\underset{i=1}{\sum }}{\left(y_{i}-\hat{y_{i}}\right)}^{2}\;}$$

### 5.2. Data Packet Loss Rate Test

To verify the reliability of the network layer of the system, the packet loss rate of the wireless transmission network is tested at different communication distances. With the IoT intelligent monitoring system connected to the server and router, the mobile collection nodes send 500 data packets at intervals of 50 m, 70 m, 90 m, 110 m, 130 m, and 150 m for testing the packet loss rate. From Table 3, it can be concluded that this system can achieve reliable data communication within the communication distance of 90 m and can meet the application requirements.

### 5.3. System Stability Test

To verify the stability of this system, the water quality, environment, and plant growth conditions of the aquaponics system were monitored for 24 h of continuous operation. The real-time monitoring and historical data page of the Mini Program are shown in Figure 12.

Figure 12 represents the system that can collect lots of key indicators of the aquaponics system stably, without disconnection during the monitoring process. Meanwhile, it can accomplish functions of data wireless transmission, trend curve drawing, and database storage, which indicates that the system is stable and reliable in operation.

### 5.4. Field Test

Field testing began on 20 July 2022 and lasted for two months. The site was located in an aquaponics system within the College of Information and Electrical Engineering building at the China Agricultural University. The monitoring system was powered up and calibrated prior to the test, and the monitoring interval was set to 1 min. The system was set to take average data every 1 h and upload it to the cloud for historical storage. During the test period, the three test indicators were manually sampled at 9:00, 12:00, 15:00, 18:00, and 21:00 each day using a handheld photometer, DO detection device, and a scale respectively. The field test environment is shown in Figure 13.

Figure 14 shows the changes in DO over 3 days. Since the system is equipped with an oxygenation pump that can be intelligently controlled, the DO data as shown oscillates between 7 and 8, providing a good growing environment for the fish. Table 4 compares the indicators obtained by manual sampling with those obtained by system monitoring. The results show that the MAE of the DO value is 0.093. The MSE and RMSE is 0.012, and 0.111, respectively.

Figure 15 shows the variation of lightness over 3 days, with data cycling from large to small, consistent with the daily regular pattern. Due to the additional fill light during 2:00–5:00, the period of the graph shows a boost of data. Table 5 compares the indicators obtained by manual sampling with those obtained by systematic monitoring. The results show that the MAE of light intensity is 0.045. The MSE and RMSE is 0.007 and 0.082, respectively.

The validation of the growth monitoring module lasted for one month due to the slow growth rate of the plants. Figure 16 shows the change in plant height during this period with an increasing trend. Table 6 compares the indicators obtained by manual sampling with those obtained by systematic monitoring using edge-computing. The results show that the MAE of plant height value is 0.758. The MSE and RMSE is 0.583 and 0.763, respectively. Though, it is clear that there is a uniform bias due to image acquisition calibration. Thus, the result is helpful to evaluate plant growth with tiny data transmission.

In summary, the embedded edge-computing-based aquaponics network monitoring system can effectively monitor various key indicators in the aquaponics system and provide valuable references for agricultural production. Especially for plant growth monitoring, the use of edge sensors for image processing makes the transmission efficiency much higher. 

## 6. Conclusions

As this study has demonstrated, we focused on the integrity of monitoring parameters in the aquaponics system and realized the possibility of simultaneous remote monitoring and intelligent control of the environment, water quality, and plant growth conditions. The proposed system ensures the quality operation of aquaponics system and lays a foundation for the research of cultivation-breeding ratio to a certain extent.

As for the environment and water-quality parameters, the system overcomes many problems of the traditional handheld aquaponics system, such as high latency, labor-intensive, low efficiency, and poor scalability. Furthermore, in order to monitor the growth conditions of plants in the aquaponics system and improve the intelligence of the system, edge computing was introduced to build an end-edge-cloud system architecture. Thus, deep learning algorithms can be used for image processing. In this way, the computing tasks of the traditional system are transferred from the data center to the edge sensor, realizing the diversion of high energy-consuming image data and low energy-consuming sensing data. It proves that the edge sensor can improve the timeliness of data calculation, reduce the flow pressure, improve the efficiency of data transmission and enhance the scalability of the system. Using the WeChat Mini Program as the software platform, users can easily access remote monitoring and control of the aquaponics system by using a smart cell phone. The test results show that the system can provide stable local data collection and remote transmission, trend curve plotting, and database storage services for aquaponics systems effectively. The packet loss rate of the wireless transmission network is 0% within 90 m and 3.2% at 110 m. The system measurement data is proven to reflect the system condition well.

Due to the limited site, the research object of this paper is set in the miniaturized aquaponics system. In theory, a larger site can be covered by more monitoring nodes. This paper provides a good basis for further discussion and study. In the future, further research can make the system more efficient and intelligent by strengthening the computing power at the edge, such as further pruning and compression of the model or connecting the accelerator for deep learning in the periphery of the microcontroller. 

## Figures and Tables

**Figure 1 sensors-22-09260-f001:**
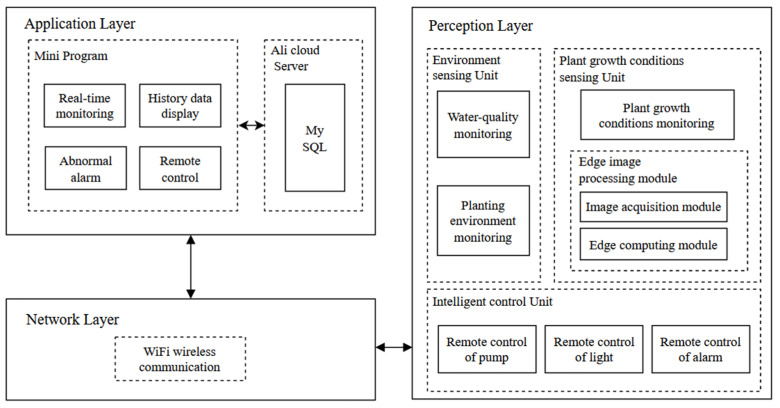
The general framework of the system.

**Figure 2 sensors-22-09260-f002:**
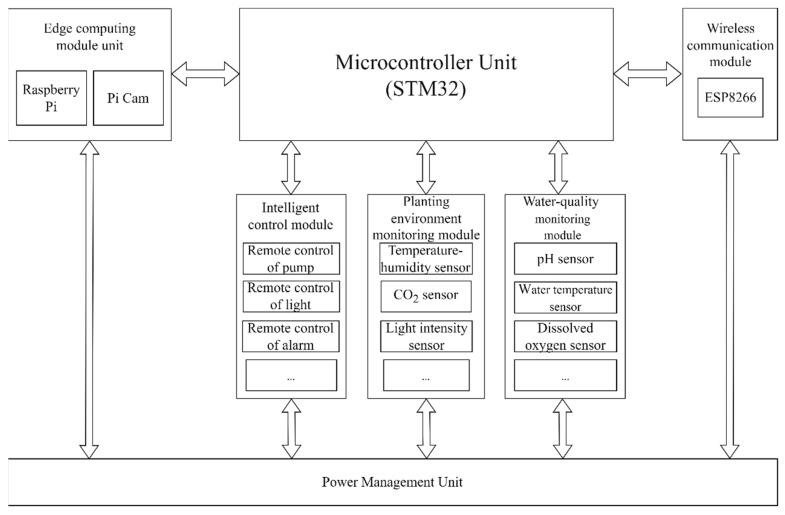
The general framework of the hardware.

**Figure 3 sensors-22-09260-f003:**
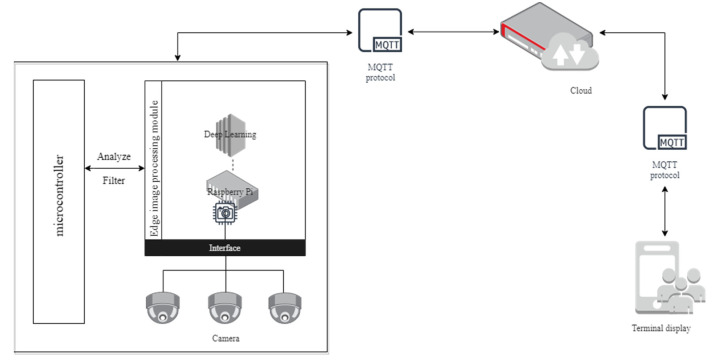
The architecture of edge image processing module.

**Figure 4 sensors-22-09260-f004:**
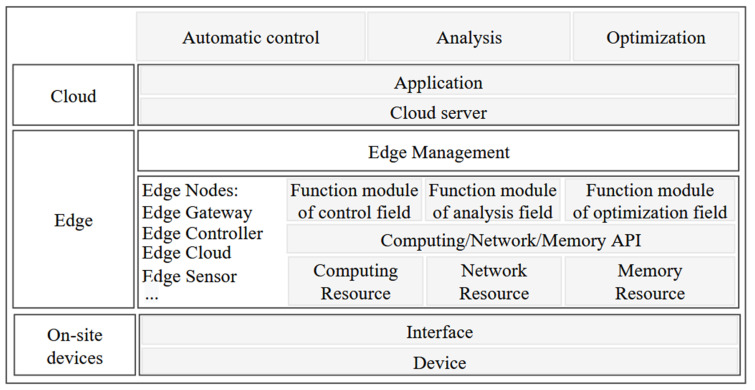
The basic network architecture of edge-computing.

**Figure 5 sensors-22-09260-f005:**
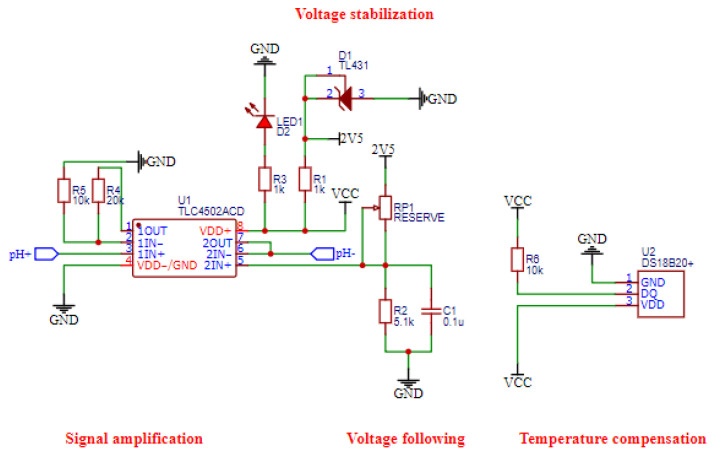
Data acquisition circuit: The number stands for the pin.

**Figure 6 sensors-22-09260-f006:**
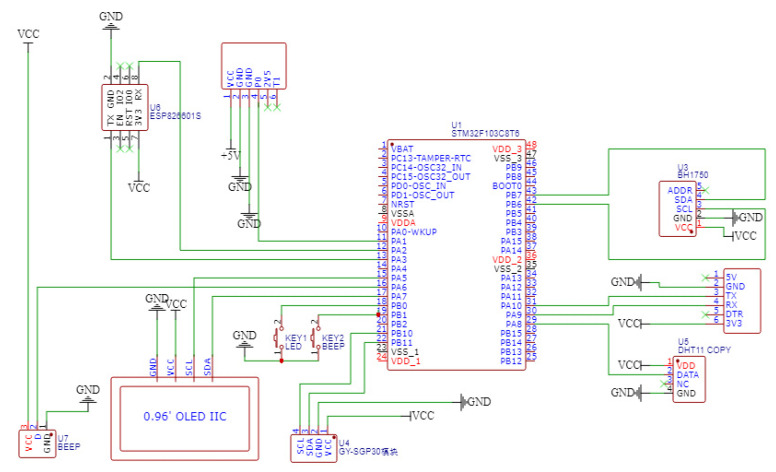
Main control circuit and peripheral interaction circuit: The number stands for the pin.

**Figure 7 sensors-22-09260-f007:**
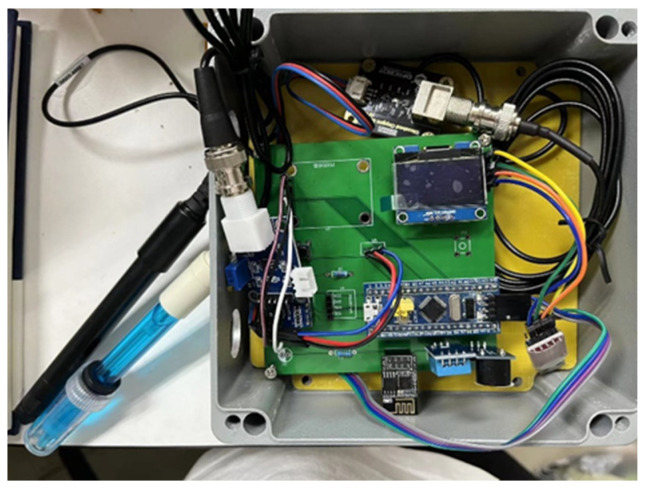
Physical picture of data acquisition node and transceiver.

**Figure 8 sensors-22-09260-f008:**
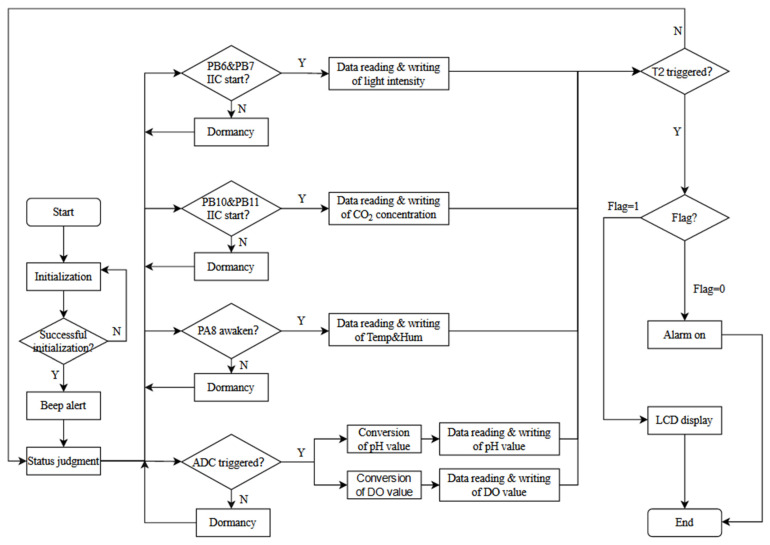
Block program diagram of intelligent sensing unit.

**Figure 9 sensors-22-09260-f009:**
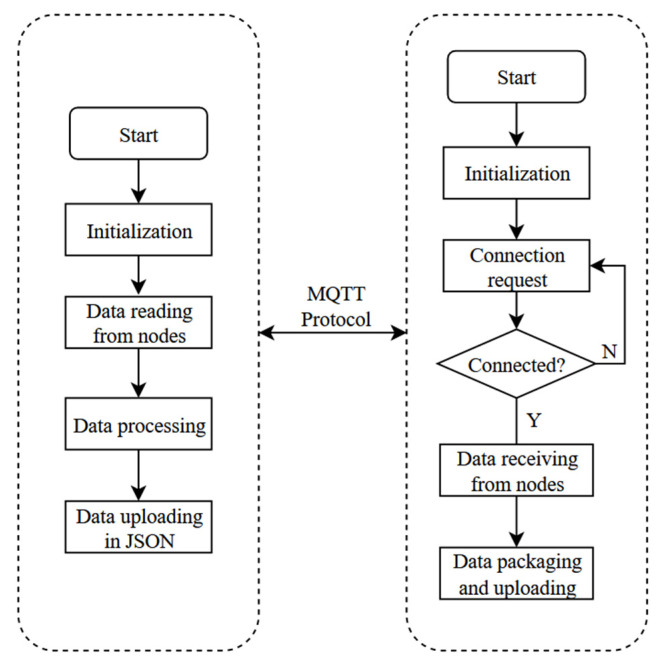
Block program diagram of wireless transmission network.

**Figure 10 sensors-22-09260-f010:**
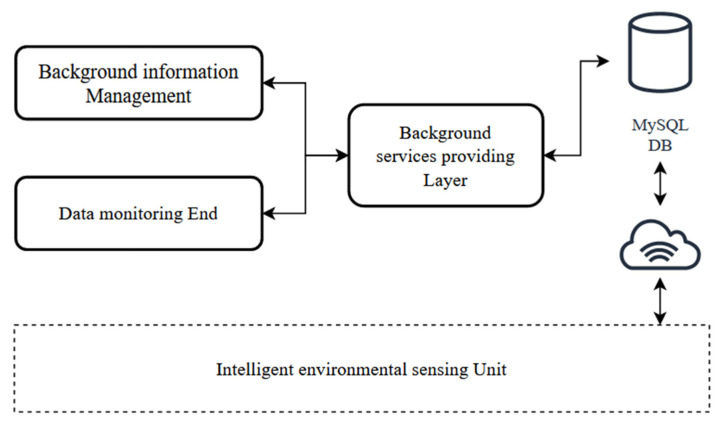
The architecture of the application layer.

**Figure 11 sensors-22-09260-f011:**
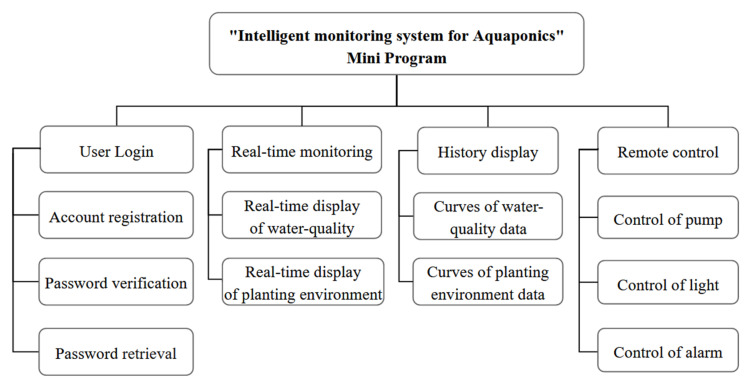
Block diagram of the Mini Program function.

**Figure 12 sensors-22-09260-f012:**
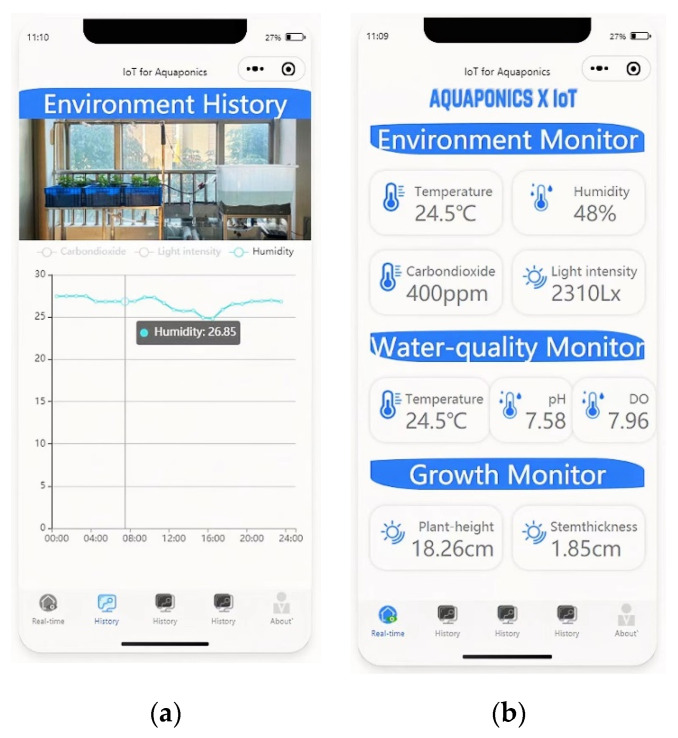
Pages of Mini Program for monitoring aquaponics system. (**a**) The curves of 3 environment indicators in 24 h; (**b**) Page of real-time system monitoring.

**Figure 13 sensors-22-09260-f013:**
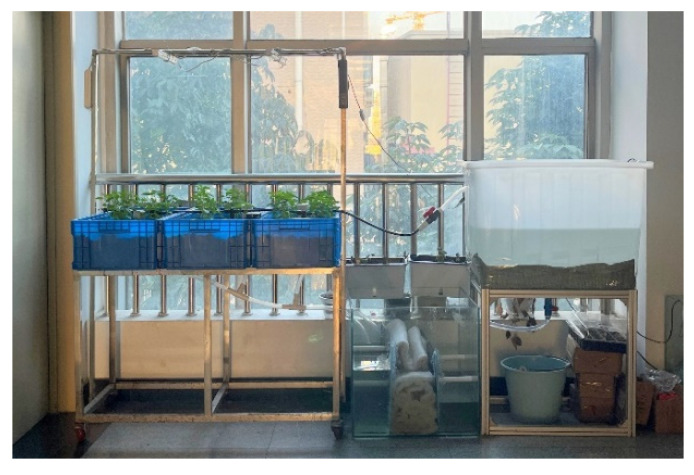
Proposed aquaponics system.

**Figure 14 sensors-22-09260-f014:**
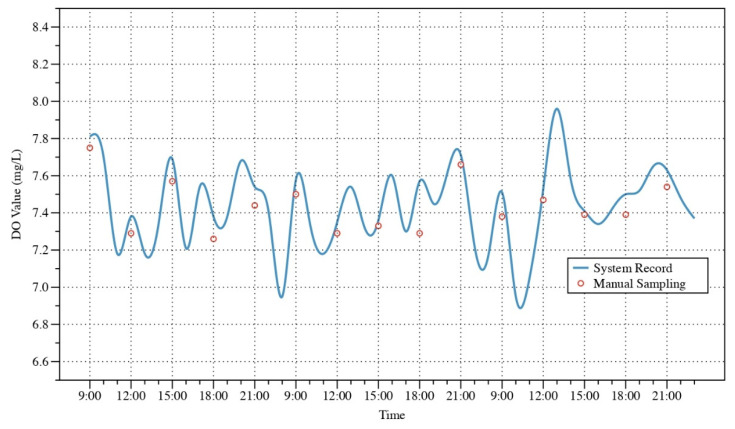
System record value and manual sampling of DO.

**Figure 15 sensors-22-09260-f015:**
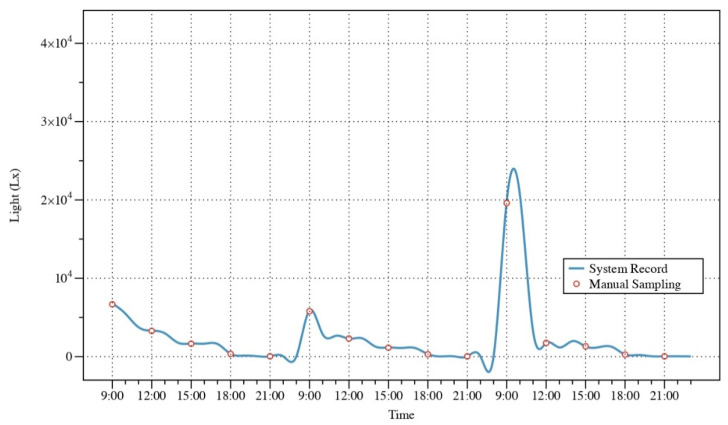
System record value and manual sampling of light intensity.

**Figure 16 sensors-22-09260-f016:**
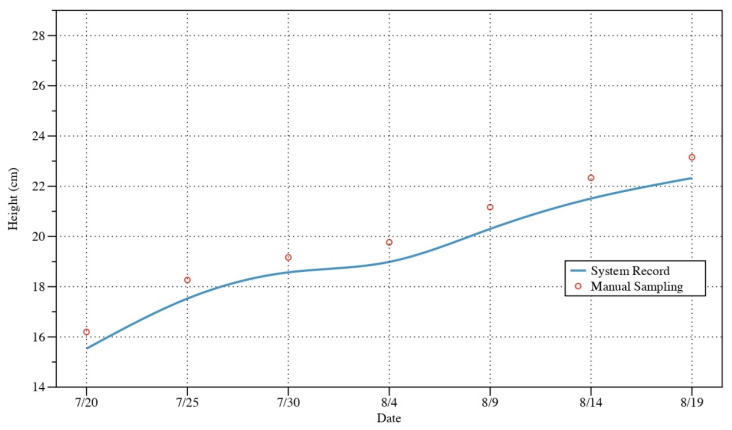
System record value and manual sampling of plant height.

**Table 1 sensors-22-09260-t001:** Parameters of sensors.

Sensors	Measurement Range	Output Type	Precision
SGP30	0~60,000 ppm	I^2^C	15%
DHT11	TEM: 0~50 °C; HUM: 20~90% RH	1-Wire^®^	TEM: ±1%; HUM: ±4%
BH1750	1~65535 Lx	I^2^C	±20%
DS18B20	55~125 °C	1-Wire^®^	±0.5 °C

**Table 2 sensors-22-09260-t002:** AT instruction set of ESP8266.

Description	Instruction	Response
Test	AT	OK
Reset (restart)	AT + RST	OK
Set baud rate	AT + CIOBAUD = BaudRate	OK
Set working mode	AT + CWMODE = Mode	OK
Set the pass-through mode	AT + CIPMODE = Mode	OK
Query the connected AP currently	AT + CWJAP?	Current AP Information
Connecting to Hotspots	AT + CWJAP = “SSID”, “password”	OK
Set the single connection mode	AT + CIPMUX = 0	OK
Establishing a TCP connection	AT + CIPSTART = “TCP”, XXXX	OK
Transmission of data	AT + CIPSEND	OK

**Table 3 sensors-22-09260-t003:** Packet loss rate of the wireless transmission network.

FN	DIS/m	NOP/pc	NORP/pc	PLR/%
1	50	500	500	0
2	70	500	500	0
3	90	500	500	0
4	110	500	484	3.2
5	150	500	478	4.4
6	200	500	455	9

**Table 4 sensors-22-09260-t004:** Comparison of the results of DO value.

FN	Date	Time	System Record	Manual Sampling	MAE	MSE	RMSE
1	7/25	09:00	7.81	7.75	0.093	0.012	0.111
2	7/25	12:00	7.38	7.29			
3	7/25	15:00	7.69	7.57			
4	7/25	18:00	7.38	7.26			
5	7/25	21:00	7.54	7.44			
6	7/26	09:00	7.58	7.50			
7	7/26	12:00	7.35	7.29			
8	7/26	15:00	7.36	7.33			
9	7/26	18:00	7.57	7.29			
10	7/26	21:00	7.71	7.66			
11	7/27	09:00	7.51	7.38			
12	7/27	12:00	7.53	7.47			
13	7/27	15:00	7.41	7.39			
14	7/27	18:00	7.50	7.39			
15	7/27	21:00	7.63	7.54			

**Table 5 sensors-22-09260-t005:** Comparison of the results of light intensity value.

FN	Date	Time	System Record/10^3^ Lx	Manual Sampling/10^3^ Lx	MAE	MSE	RMSE
1	9/10	09:00	6.799	6.680	0.045		0.082
2	9/10	12:00	3.312	3.280			
3	9/10	15:00	1.678	1.650			
4	9/10	18:00	0.355	0.350			
5	9/10	21:00	0.038	0.040			
6	9/11	09:00	5.889	5.800			
7	9/11	12:00	2.289	2.310			
8	9/11	15:00	1.173	1.140		0.007	
9	9/11	18:00	0.320	0.300			
10	9/11	21:00	0.035	0.033			
11	9/12	09:00	19.870	19.600			
12	9/12	12:00	1.733	1.750			
13	9/12	15:00	1.295	1.320			
14	9/12	18:00	0.268	0.259			
15	9/12	21:00	0.041	0.038			

**Table 6 sensors-22-09260-t006:** Comparison of the results of plant height value.

FN	Date	System Record	Manual Sampling	MAE	MSE	RMSE
1	7/20	15.53	16.20	0.758	0.583	0.763
2	7/25	17.53	18.27			
3	7/30	18.57	19.17			
4	8/4	18.99	19.77			
5	8/9	20.30	21.17			
6	8/14	21.51	22.33			
7	8/19	22.32	23.15			

## Data Availability

Not applicable.

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
