# Peer review of "A Modularized IoT Monitoring System with Edge-Computing for Aquaponics"

_sensors, 2022, doi:10.3390/s22239260_

Round 1

Reviewer 1 Report

The paper deals with a very interesting topic for modern smart agriculture.

In addition to monitoring environmental parameters, plant growth is considered through deep learning algorithms in an edge-computing environment.

The system architecture is based on standard components. No neural accelerator is considered to speed up the operations of neural networks.

The limit of experimentation is having considered only one node for edge computing and not have described what happens in large plants when more nodes are needed.

The results shown are satisfactory, and the approach is the overall sound.

Author Response

Dear Reviewer:

Please see the attachment. The point-by-point response is on Page 26.

Thanks for your time.

Best wishes

Yours sincerely

Wang Haihua

Reviewer 2 Report

- The abstract need to be rewritten, lines 13-17 is a too long sentence. Also, in the abstract, the authors mention the lack of mature technical standard, but their paper is not about developing a standard. Their motivation need to be properly articulated. In addition, avoid using undefined abbreviations in the abstract as it needs to be self contained. 

- Define abbreviations on first reference.

- The reference to figure need to be consistent and according to the journal template, sometime it is mentioned as Figure1 and in others as Figure 2 and figure 2.

- Line 141-142, improve the English.

- Figure 7 is not a diagram, it is a picture. 

- The quality of figures need to be improved. The text in the figures is pale.

- Define the metrics in Table 4.

- The conclusion need to be better formatted. 

- The table of abbreviations is missing.

- A critical review of the literature is missing and how this study is different. 

- Line 90, improper sentence, rephrase.

- Line 96,  "General" --> "general"

Author Response

Dear Reviewer:

Please see the attachment. The point-by-point response is on Page 26-28.

Thanks for your time.

Best wishes

Yours sincerely

Wang Haihua 

Reviewer 3 Report

This paper provides the edge-computing based Internet of Things (IoT) technology to overcome many problems of traditional handheld aquaponics system, such as high latency, labor-intensive, low efficiency and poor scalability. Thanks to the software platform of the WeChat Mini Program, users can easily monitor and control the aquaponics system remotely using their smart phones. The paper is timely, well-handled and very well organized. However, the following issues need to be addressed before acceptance.

1.     The Introduction section should emphasize the main contributions and structure of the paper.

2.     The last paragraph of the Introduction section should mention the organization of the rest of the paper.

3.     It is important for the authors to include some of the recently proposed (2018-2022) cutting-edge (state-of-the-art) studies. It is useful that more than half of the references used are between 2018-2022.

a.     Taha, M. F., ElMasry, G., Gouda, M., Zhou, L., Liang, N., Abdalla, A., ... & Qiu, Z. (2022). Recent Advances of Smart Systems and Internet of Things (IoT) for Aquaponics Automation: A Comprehensive Overview. Chemosensors, 10(8), 303.

b.     Ulum, M., Ibadillah, A. F., Alfita, R., Aji, K., & Rizkyandi, R. (2019, April). Smart aquaponic system based Internet of Things (IoT). In Journal of Physics: Conference Series (Vol. 1211, No. 1, p. 012047). IOP Publishing.

c.     Ezzahoui, I., Abdelouahid, R. A., Taji, K., & Marzak, A. (2021). Hydroponic and Aquaponic Farming: Comparative Study Based on Internet of things IoT technologies. Procedia Computer Science, 191, 499-504.

d.     Khaoula, T., Abdelouahid, R. A., Ezzahoui, I., & Marzak, A. (2021). Architecture design of monitoring and controlling of IoT-based aquaponics system powered by solar energy. Procedia Computer Science, 191, 493-498.

e.     Alselek, M., Alcaraz-Calero, J. M., Segura-Garcia, J., & Wang, Q. (2022). Water IoT Monitoring System for Aquaponics Health and Fishery Applications. Sensors22(19), 7679.

4.     On 1st page (abstract section) line 19, MCU acronym first appears here, authors should firstly introduce microcontroller (MCU), then you can use the term thereafter.  

5.     On page 2, line 47, the abbreviation 'DO' is used, it is useful to write the definition in first appearance. (DO, dissolved oxygen). Please follow the academic writing standards.

6.     Why were the data from different sensors not taken at the same time? (plant height, 7/17...8/21; DO, 7/25/7/27, intensity value, 9/10....9/12) Any particular reason?

Author Response

Dear Reviewer:

Please see the attachment. The point-by-point response is on Page 26-27.

Thanks for your time.

Best wishes

Yours sincerely

Wang Haihua 

Round 2

Reviewer 2 Report

The authors addressed most of my comments. 

Reviewer 3 Report

The authors have progressed in improving the paper (sensors-2026270-peer-review-v2). compared to previous versions of the paper(sensors-2026270-peer-review-v1).  When both versions are evaluated comparatively, it is seen that the authors make the corrections requested by the referees and show the necessary sensitivity in the revision of the paper in line with the comments. In the revised version of the paper, almost all the comments have been considered and addressed by the authors.

The response to reviewers file is well-prepared. The changes made by the authors in line with the opinions/suggestions/evaluations of the referees can be tracked. The existing organization and spelling problems in the previous version of the paper have been fixed. In the revised version, the clarity and follow-up of the study have been increased. In addition, the article has been carefully reviewed for grammatical and typos.

As a result, my concerns on the previous version of the paper have disappeared with the explanations made by the authors, as well as the revision they have made.

This revision is sufficient, and it is possible to evaluate the paper for publication after preparation according to MDPI Journal template.